# Gestational Vitamin E Status and Gestational Diabetes Mellitus: A Retrospective Cohort Study

**DOI:** 10.3390/nu15071598

**Published:** 2023-03-25

**Authors:** Huifeng Shi, Xiaoli Gong, Qing Sheng, Xiang Li, Ying Wang, Tianchen Wu, Yangyu Zhao, Yuan Wei

**Affiliations:** 1Department of Obstetrics and Gynecology, Peking University Third Hospital, Beijing 100191, China; nsxm@pku.edu.cn (H.S.); 19801299010@163.com (X.G.);; 2National Clinical Research Centre for Obstetrics and Gynecology, Beijing 100191, China; 3National Centre for Healthcare Quality Management in Obstetrics, Beijing 100191, China

**Keywords:** vitamin E, gestational diabetes mellitus, cohort study

## Abstract

Objectives: To examine the association between vitamin E (VE) status and gestational diabetes mellitus (GDM). Methods: A retrospective cohort study was conducted by using data of 52,791 women at 137 hospitals across 22 provinces of China. A fasting plasma glucose (FPG) level of ≥5.1 mmol/L between the 24th and 40th weeks of gestation was used as the criteria for the diagnosis of GDM. Mean FPG level and GDM rate were calculated within each combination of the first-trimester VE concentration categories and gestational change categories. The associations of the first-trimester VE concentrations and gestational VE change with FPG and GDM were examined by employing generalized additive models (GAMs). Results: 7162 (13.57%) cases were diagnosed with GDM. The GDM rate was 22.44%, 11.50%, 13.41%, 12.87%, 13.17%, 13.44%, 12.64%, and 14.24% among women with the first-trimester VE concentrations of <7.2, 7.2–7.9, 8.0–9.3, 9.4–11.0, 11.1–13.2, 13.3–15.8, 15.9–17.7, and 17.8–35.9 mg/L, respectively. The GDM rate was 15.96%, 13.10%, 13.64%, and 12.87% among women with gestational VE change of <0, 0–0.19, 0.20–0.29, ≥0.30 mg/L per week, respectively. Multivariable adjusted GAM analyses found that the first-trimester VE concentration was associated with the FPG levels and GDM risk in an L-shaped pattern; the FPG levels and GDM risk decreased sharply to a threshold (around 7 mg/L), and then were keep flat. Gestational VE decreases when the first-trimester VE level was less than 11 mg/L were related to increased FPG levels and GDM risk. Conclusions: Both low first-trimester VE levels and subsequent gestational VE decrease were related with increased risk of GDM. The findings suggest the necessity of having VE-rich foods and appropriate VE supplementation to prevent GDM for pregnant women with low baseline VE levels.

## 1. Introduction

Gestational diabetes mellitus (GDM) is one of the most common complications in pregnancy. The prevalence of GDM is 1.8–25% in different countries and areas [1], and it was estimated to be 14.8% in China [2]. Women with GDM have higher risk of preeclampsia, cesarean section, type 2 diabetes after pregnancy, and fetal adverse outcomes, including macrosomia, shoulder dystocia, birth trauma, and neonatal hypoglycemia [3].

Oxidative stress may be implicated in the pathogenesis of diabetes mellitus and its complications [4,5,6,7,8]. The underlying pathophysiology is similar to type 2 diabetes in most instances, which is why hyperglycemia leads to excessive production of intracellular reactive oxygen species (ROS) with insufficient clearance, thus enhancing the oxidative stress level of the body, leading to adverse pregnancy outcomes [4,5,6,7]. Vitamin E (VE), a well documented antioxidant, can neutralize ROS, eliminate oxygen free radicals, and protect tissue from damage [9,10]. However, although a robust number of studies on the role of VE in the cause and prevention of diabetic complications [8,11,12,13,14,15], there are very few studies on its association with risk of diabetes mellitus, especially GDM [16,17]. The systematic review by Sharifipour et al. aimed to evaluate the association between VE and GDM, and this only included eleven relevant studies with a total of 596 participants published before December 2019, of which seven were the case-control studies, and the rest were cross-sectional [18]. Our further search of the literature identified very few relevant articles published since then. The biggest hurdle is the cost and labor involved in obtaining a large enough sample size to examine the quantitative association, given the low proportion of VE deficiency. More studies are evidently needed on this topic, especially those with longitudinal design.

In response to the lack of research, we did this retrospective large-sample cohort study to examine the association between VE and GDM.

## 2. Materials and Methods

### 2.1. Study Design and Population

An original prospective cohort study investigating VE concentrations was implemented at 180 hospitals across 23 provinces of China during 2015–2019. Pregnant women who planned to be registered for antenatal care in these hospitals were recruited to participate, and blood samples were collected during antenatal care visits.

In this present study, we did a retrospective analysis by using data from the prospective cohort. We included singleton pregnancies with fasting plasma glucose (FPG) test and available data of VE concentrations at gestational age of 4–13 and 24–40 weeks. Women with prepregnancy diabetes diagnosis or FPG of ≥7.0 mmol/L and those with age of less than 15 or more than 49 were excluded. Finally, we included 52,791 women at 137 hospitals across 22 provinces (Anhui, Beijing, Guangdong, Guangxi, Hebei, Henan, Heilongjiang, Hubei, Hunan, Jilin, Jiangsu, Liaoning, Inner Mongolia, Qinghai, Shandong, Shanxi, Shaanxi, Sichuan, Tianjin, Yunnan, Zhejiang, and Chongqing). All study process was reviewed and approved by the Peking University Third Hospital Medical Science Research Ethics Committee (IRB00006761–2015277).

### 2.2. Gestational VE Status

A fasting venous blood sample was collected by trained nurse technicians when the pregnant woman went on regular antenatal care. Laboratory staff centrifuged the samples within 1 h, extracted and aliquoted serum immediately under no-light condition, and then stored serum at −80 °C in a refrigerator. Subsequently, all serum samples were sent to the only collaborating laboratory on dry ice within 1 month and maintained at −80 °C until analyzed. After receiving samples, serum concentrations of VE (α-tocopherol) were quantified by high performance liquid chromatography (Prominence LC-20A, Shimadzu) in a dark room within 24 h. One-twentieth of each batch of samples were tested repeatedly, showing inter-assay coefficients of variation of <5%. The relative standard deviations were 3.86% and 4.52% within and between days, respectively.

In our analysis, we used the minimum VE concentration tested in the first trimester and the VE concentration at the time of FPG test with the highest FPG value in 24–40 weeks. We calculated gestational VE change by subtracting the first-trimester VE concentration from the VE concentration at the FPG test time and then dividing the results after subtraction by the interval of gestational weeks. This is a reasonable approach because there was no previously-published data of gestational VE gain reference charts for Chinese population to calculating gestational age-standardized z scores, and our data show a linear trend of VE gain during pregnancy.

### 2.3. Fasting Plasma Glucose Level and Diagnosis on Gestational Diabetes Mellitus

We extracted the highest fasting plasma glucose values at gestational age of 4–13 and 24–40 weeks from the original cohort data collected from electronic medical records. In pregnant women with no prepregnancy diabetes diagnosis and FPG of less than 7.0 mmol/L, a fasting plasma glucose level of ≥5.1 mmol/L between the 24th and 40th weeks of gestation was used as the criteria for the diagnosis of GDM [19].

### 2.4. Covariates

Information about sociodemographic characteristics and medical conditions was collected from medical records, including geographical region, ethnic origin, education, household registration (urban residents, rural residents, or rural-to-urban migrants), age, mode of conception, primigravida, pre-pregnancy body mass index (BMI), and hypertension.

### 2.5. Statistical Analysis

We calculated proportions to describe the distribution of geographical region, ethnic origin, education, household registration, age, mode of conception, primigravida, pre-pregnancy BMI group, and hypertension, according to the GDM status. Mean FPG level and GDM rate were calculated within each combination of the first-trimester VE concentration categories (<7.2, 7.2–7.9, 8.0–9.3, 9.4–11.0, 11.1–13.2, 13.3–15.8, 15.9–17.7, and 17.8–35.9 mg/L; divided according to the 5th, 10th, 25th, 50th, 75th, 90th, and 95th percentiles) and gestational VE change categories (<0, 0–0.19, 0.20–0.29, ≥0.30 mg/L per week; divided according to the 25th, 50th, and 75th percentiles).

The associations of the first-trimester VE concentration and the gestational VE change with FPG and GDM were identified by performing generalized additive models (GAMs) with smooth functions implemented in the Mgcv version 1.8 package in R statistical software. All models were adjusted for the nine covariates and FPG levels at gestational age of 4–13 weeks, and the confounding of the first-trimester VE concentration and the gestational VE change on each other. According to the models, predicted concentration of FPG and risk of GDM with 95% confidence intervals (CIs) were calculated with respect to the first-trimester VE concentration and the gestational VE change. Subsequently, we used the visreg 2.7.0.3 package for visualizing these regression models. Heatmaps were constructed to exhibit the differences based on combinations of the first-trimester VE concentration and the gestational VE change (red represents high FPG level or GDM incidence, blue represents low FPG level or GDM incidence).

Furthermore, we used a generalized linear model (GLM) to assess the association of the first-trimester VE concentration categories and gestational VE change categories with FPG and used robust Poisson regression to assess the associations with GDM. Sensitivity analyses were performed by adjusting for different covariates. Model A was adjusted for the nine covariates and FPG levels at a gestational age of 4–13 weeks. Model B was additionally adjusted for the confounding of the first-trimester VE concentration categories and the gestational VE change categories on each other. Model C was additionally adjusted for an interaction term of the first-trimester VE concentration categories and the gestational VE change categories based on Model B. We also performed a stratified analysis to examine the association of gestational VE change categories with FPG and GDM in each first-trimester VE concentration category, by using robust Poisson regression models that adjusted for the aforementioned nine covariates. Regression coefficient (β) or relative risks (RRs) with 95% CIs were calculated.

We performed statistical analyses with Statistical Package for the Social Sciences (SPSS) software 25.0 (SPSS, Inc., Chicago, IL, USA) and the R statistical software, version 3.6.2. A two-tailed *p*-value < 0.05 was considered statistically significant.

## 3. Results

A total of 7162 (13.57%) GDM cases were identified among 52,791 pregnant women (mean age [SD], 28.87 [4.33] years). The baseline characteristics of participants are detailed in Table 1.

Mean FPG level and GDM rate within the combination of the first-trimester VE concentration categories and gestational VE change categories are presented in Figure 1. The mean FPG concentrations were 4.78, 4.63, 4.62, 4.61, 4.60, 4.60, 4.60, and 4.65 mmol/L, and the GDM rates were 22.44%, 11.50%, 13.41%, 12.87%, 13.17%, 13.44%, 12.64%, and 14.24% among women with first-trimester VE concentrations of <7.2 (<5th), 7.2–7.9 (5th–9th), 8.0–9.3 (10th–24th), 9.4–11.0 (25th–49th),11.1–13.2 (50th–74th), 13.3–15.8 (75th–89th), 15.9–17.7 (90th–95th), and 17.8–35.9 (95th–100th) mg/L, respectively. The mean FPG concentrations were 4.70, 4.60, 4.61, and 4.61 mmol/L, and the GDM rates were 15.96%, 13.10%, 13.64%, and 12.87% among women with gestational VE change of <0 (<25th), 0–0.19 (25th–49th), 0.20–0.29 (50th–74th), ≥0.30 (75th–100th) mg/L per week, respectively. Higher FPG levels and GDM risks were showed in those combinations of the lower first-trimester VE concentration and gestational VE decrease categories.

Multivariable adjusted GAM analyses showed that the first-trimester VE concentration was related to FPG levels (Figure 2a) and GDM risk (Figure 2b) in a L-shaped pattern; the FPG levels and GDM risk decreased sharply to a threshold (around 7 mg/L), and then they were kept flat. The association of gestational VE change to the FPG levels is presented as a reverse J-shaped curve, and the GDM risk is presented as a declining curve. Gestational VE decrease was associated with increased FPG levels and GDM risk (Figure 2c,d). Considerable differences in the FPG levels and GDM risk according to the combination of the first-trimester VE concentration and gestational VE change were identified (Figure 3). Higher FPG levels and GDM risks were found in women with lower first-trimester VE concentration and greater gestational VE decrease.

The results of multivariable adjusted robust Poisson regression on the association of gestational VE status with FPG and GDM was presented in Table 2. Compared to women with first-trimester VE concentrations of 11.1–13.2 (50th–74th) mg/L, those with first-trimester VE concentrations of <7.2 (<5th) mg/L had 0.123 (95% CI: 0.081, 0.166) increase in FPG and 1.36-fold (95% CI: 1.14, 1.62) increased risks for GDM. Gestational VE decrease was related with 0.050 (95% CI: 0.019, 0.082) increase in FPG, while gestational vitamin E gain of ≥0.30 mg/L per week was related to a −0.040 (95% CI: −0.063, −0.017) decrease in FPG. Sensitivity analyses showed that the estimates were robust. The stratified analysis showed that within the first-trimester VE concentration categories of less than 11 mg/L, gestational VE decreases were significantly associated with increased FPG levels and GDM risk (Table 3).

## 4. Discussion

To our knowledge, this is the largest cohort study to date to investigate the association between VE and GDM. Several study characteristics, including unified and well established measures for gestational VE determination and use of FPG, the laboratory test indicator for assessing GDM, were beneficial for our analysis, as they minimized recall bias due to the retrospective nature. We found that the low first-trimester VE level and subsequent gestational VE decrease were related with increased risk of GDM, which strengthened the perception for occurrence of GDM and potential application value.

All available previous studies conducted the analyses based on the hypothesis that the association between VE concentration and GDM was linear [17]. The aforementioned systematic review by Sharifipour et al. concluded that the VE level was significantly lower in women with GDM compared to healthy pregnant women [18]. However, one large retrospective cohort study (*n* = 19,647) from China reported that maternal VE concentrations in the first and second trimesters were positively associated with GDM; this study did not exclude pregnant women with pre-gestational diabetes mellitus [16]. In contrast to previous findings, we found that the association of the first-trimester VE concentration with the GDM risk are presented as a L-shaped curve with an inflexion point around 7 mg/L. The threshold concentration identified far exceeded the clinical definition (5.0 mg/L) of VE deficiency, indicating the expansion of high-risk population in pregnancy. Large sample size, rigorous measurement, and the use of GAM models strongly contributed to the validity of our findings.

We found that maternal VE level rises in an almost linear trend during pregnancy, from 10 mg/L at 4th week to 16 mg/L at 40th week, with an average weekly increase of 0.16 mg/L. It was consistent with previous findings that VE levels increase with gestational age, which even exceed the normal level [20,21,22]. The variation may relate to the changes in lipid metabolism during pregnancy, since VE levels are highly correlated with overall fat content [22,23,24]. An animal experiment showed that dams with GDM had obvious changes in lipid metabolism, even earlier than changes in glucose [24]. In women with the first-trimester VE concentration of lower than 11 mg/L, we found that gestational VE decrease was related to increased GDM risk, whereas excessive gain showed no potential benefits in reducing the risk of GDM. These findings suggest the necessity of having VE-rich foods and appropriate VE supplementation to prevent GDM for pregnant women with low baseline vitamin E levels, although the idea needs further confirmation [25].

Our study has practical clinical application value. We recommend keeping a VE level of higher than 7 mg/L in the first trimester and avoiding gestational VE decrease when the first-trimester VE level is less than 11 mg/L to prevent GDM and other pregnancy complications, such as preeclampsia [26,27]. However, we could not recommend all pregnant women to test VE in the first trimester when the cost-effectiveness evaluation was insufficient. Therefore, development of the simple methods to identify women with low VE levels are warranted for future research. Considering the ethical issues of setting controls in RCTS, more real-world studies are needed to confirm the efficacy of VE supplementation among pregnant women with low VE levels in preventing GDM.

However, our research has several limitations. We diagnosed GDM by FPG in 24–40 weeks. It is well recognized that fasting hyperglycaemia represents predominantly hepatic insulin resistance, while post-prandial glucose excursions (and glucose excursions during OGTT) represent predominantly peripheral insulin resistance [28]. Hence, application of an OGTT for diagnosis of GDM, rather than just using FPG, might alter the result of the study, or at least affect some the calculated cut-off points for VE concentrations associated with an increased risk of GDM. However, our retrospective data were collected from 137 hospitals across 22 provinces of China, including low-resource settings where not all pregnant women had available access to OGTT. According to the IADPSG criteria, FPG of 4.8–5.0 mmol/L showed good test characteristics in screening for GDM [29]. Interestingly, another study found that mild hyperglycemia (FPG of 5.1–5.5 mmol/L) was more related with the increased risk of large for gestational age fetuses than GDM diagnosis by OGTT [30]. In addition, we failed to collect important confounders, such as dietary habits, nutritional supplements, and methods of glycemic control.

## 5. Conclusions

By the retrospective large-sample cohort, we found that the low first-trimester VE level of less than 7 mg/L and gestational VE decrease when the first-trimester VE level was less than 11 mg/L were related with increased FPG levels and GDM risk. These findings suggest the necessity of having VE-rich foods and appropriate VE supple-mentation to prevent GDM for pregnant women with low baseline VE levels, although the idea needs further confirmation. Further studies have to be conducted, including OGTT, a mandatory standard which has to be applied in all women, and it should also be used in lower standard hospitals in China in the future.

## Figures and Tables

**Figure 1 nutrients-15-01598-f001:**
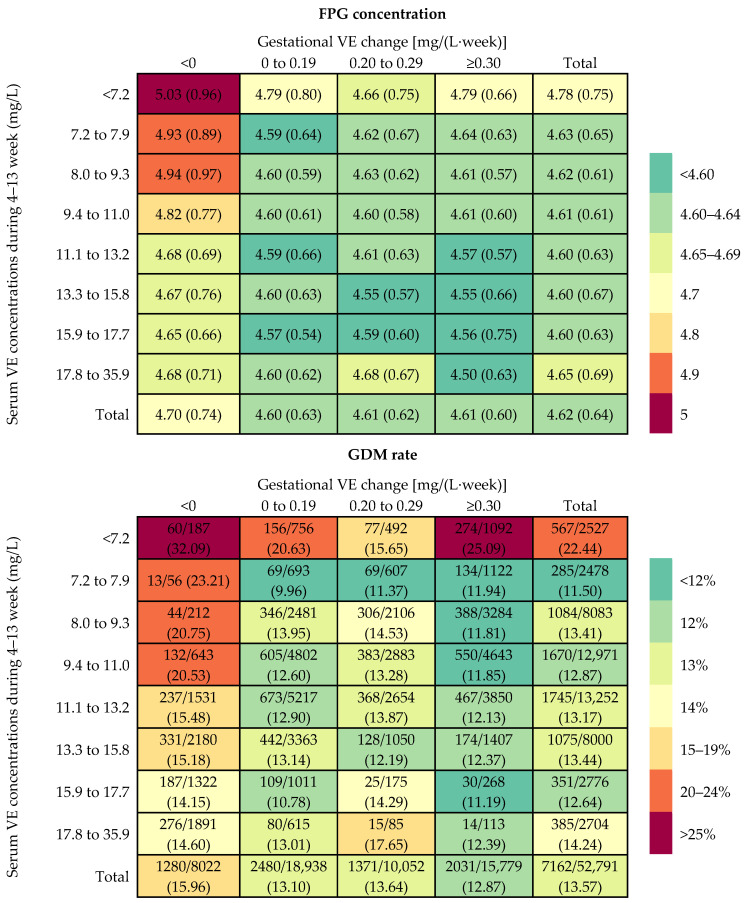
Rate of GDM and concentration of FPG within each combination of the first-trimester vitamin E (VE) concentration categories and gestational VE change categories.

**Figure 2 nutrients-15-01598-f002:**
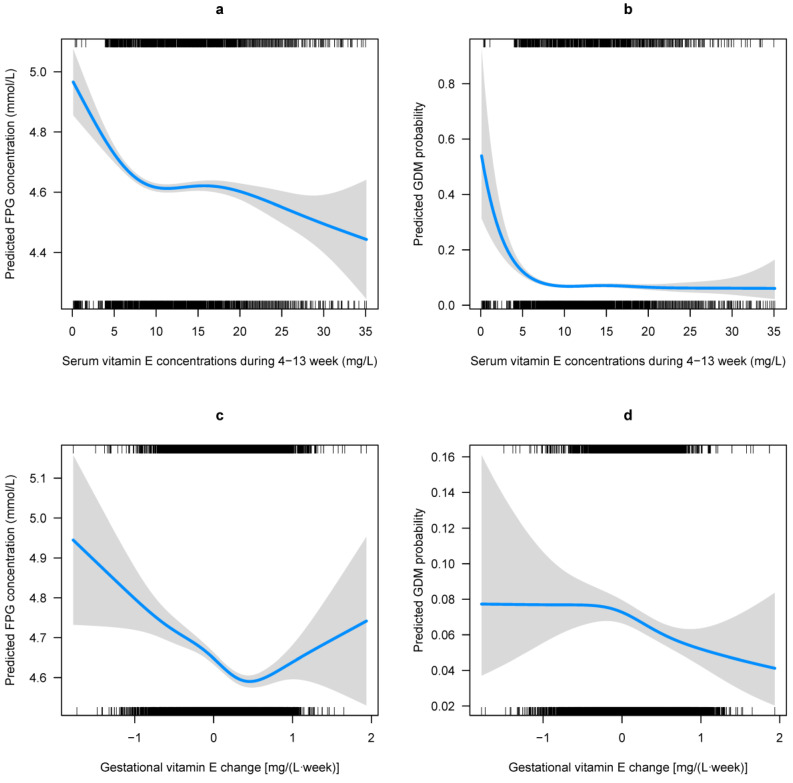
Predicted concentration of FPG and risk of GDM with respect to gestational vitamin E (VE) status. Predicted concentration of FPG and the 95% CIs were calculated with respect to (**a**) the first-trimester VE concentrations and (**c**) the gestational VE change by employing generalized additive models. Predicted risk of GDM and the 95% CIs were also calculated with respect to (**b**) the first-trimester VE concentrations and (**d**) the gestational VE change by employing generalized additive models. All models were adjusted for region, household registration, education, ethnic origin, age, ART, primigravida, pre pregnancy BMI, hypertension, and FPG levels at gestational age of 4–13 weeks, as well as the confounding of the first-trimester VE concentrations and the gestational VE change on each other.

**Figure 3 nutrients-15-01598-f003:**
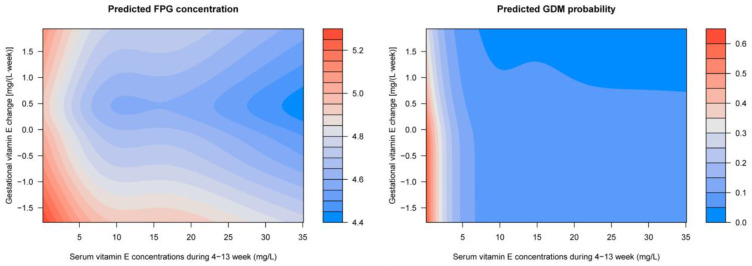
Predicted concentration of FPG and risk of GDM with respect to the first-trimester VE concentrations and gestational VE change by employing generalized additive models. Models adjusted for region, household registration, education, ethnic origin, age, ART, primigravida, pre pregnancy BMI, hypertension, and FPG levels at gestational age of 4–13 weeks, as well as the confounding of the first-trimester VE concentrations and the gestational VE change on each other.

**Table 1 nutrients-15-01598-t001:** Characteristics of participants.

	Total	Non-GDM	GDM
	(*n* = 52,791)	(*n* = 45,629)	(*n* = 7162)
Region, No. (%)			
Eastern China	29,130 (55.18)	25,805 (56.55)	3325 (46.43) *
Central China	6948 (13.16)	5530 (12.12)	1418 (19.80)
Western China	16,713 (31.66)	14,294 (31.33)	2419 (33.78)
Age, mean (SD)	28.87 (4.33)	28.85 (4.29)	28.98 (4.61)
Ethnic origin, No. (%)			
Han	51,734 (98.00)	44,698 (97.96)	7036 (98.24)
Other	1057 (2.00)	931 (2.04)	126 (1.76)
Education, No. (%)			
High school	15,446 (29.26)	12,667 (27.76)	2779 (38.80) *
College	18,285 (34.64)	16,418 (35.98)	1867 (26.07)
Master	3711 (7.03)	3479 (7.62)	232 (3.24)
Other	15,349 (29.08)	13,065 (28.63)	2284 (31.89)
ART, No. (%)			
No	52,392 (99.24)	45,309 (99.30)	7083 (98.90) *
Yes	399 (0.76)	320 (0.70)	79 (1.10)
Primigravida, No. (%)			
Yes	26,804 (50.77)	23,162 (50.76)	3642 (50.85)
No	25,987 (49.23)	22,467 (49.24)	3520 (49.15)
Household registration, No. (%)			
Urban residents	41,082 (77.82)	35,999 (78.90)	5083 (70.97) *
Migrants	3837 (7.27)	3372 (7.39)	465 (6.49)
Rural residents	7872 (14.91)	6258 (13.71)	1614 (22.54)
Hypertension, No. (%)			
No	52,513 (99.47)	45,418 (99.54)	7095 (99.06) *
Yes	278 (0.53)	211 (0.46)	67 (0.94)
Prepregnancy BMI, No. (%)			
BMI < 18.5	7080 (13.41)	6252 (13.70)	828 (11.56) *
18.5 ≤ BMI < 24	36,603 (69.34)	31,835 (69.77)	4768 (66.57)
24 ≤ BMI < 28	6908 (13.09)	5811 (12.74)	1097 (15.32)
BMI ≥ 28	1802 (3.41)	1392 (3.05)	410 (5.72)
Unknown	398 (0.75)	339 (0.74)	59 (0.82)
FPG levels (mmol/L) at 4–13 weeks, mean (SD)	4.59 (0.54)	4.52 (0.48)	5.08 (0.63) *
Vitamin E levels (mg/L) at 4–13 weeks, mean (SD)	11.61 (3.35)	11.63 (3.32)	11.48 (3.57) *

* *p* < 0.05.

**Table 2 nutrients-15-01598-t002:** Adjusted FPG concentration differences and relative risks for GDM according to the first-trimester vitamin E (VE) concentration categories and gestational VE change categories.

	FPG Concentration Differences			GDM Risk Differences		
	Model A	Model B	Model C	Model A	Model B	Model C
	β (95% CI)	β (95% CI)	β (95% CI)	RR (95% CI)	RR (95% CI)	RR (95% CI)
Serum VE concentrations during 4–13 week (mg/L)						
<7.2	**0.096 (0.072, 0.119)**	**0.105 (0.081, 0.129)**	**0.123 (0.081, 0.166)**	**1.46 (1.32, 1.60)**	**1.49 (1.36, 1.65)**	**1.36 (1.14, 1.62)**
7.2 to 7.9	−0.016 (−0.040, 0.007)	−0.004 (−0.028, 0.020)	−0.029 (−0.072, 0.015)	**0.87 (0.76, 0.98)**	0.89 (0.79, 1.01)	0.80 (0.62, 1.02)
8.0 to 9.3	−0.002 (−0.018, 0.013)	0.009 (−0.007, 0.024)	−0.012 (−0.038, 0.015)	1.01 (0.94, 1.09)	1.03 (0.96, 1.12)	1.04 (0.92, 1.19)
9.4 to 11.0	−0.001 (−0.015, 0.012)	0.006 (−0.008, 0.019)	−0.010 (−0.032, 0.012)	0.99 (0.92, 1.05)	1.00 (0.94, 1.07)	0.99 (0.88, 1.10)
11.1 to 13.2	0.00 [Reference]	0.00 [Reference]	0.00 [Reference]	1.00 [Reference]	1.00 [Reference]	1.00 [Reference]
13.3 to 15.8	**0.017 (0.002, 0.032)**	0.002 (−0.013, 0.018)	0.023 (−0.001, 0.047)	1.04 (0.96, 1.12)	1.01 (0.93, 1.09)	1.02 (0.90, 1.15)
15.9 to 17.7	**0.030 (0.007, 0.052)**	−0.002 (−0.025, 0.022)	0.012 (−0.025, 0.049)	1.00 (0.89, 1.12)	0.93 (0.83, 1.05)	0.88 (0.72, 1.08)
17.8 to 35.9	**0.040 (0.017, 0.063)**	−0.008 (−0.033, 0.016)	0.032 (−0.014, 0.078)	1.00 (0.89, 1.12)	0.90 (0.80, 1.01)	0.99 (0.78, 1.24)
Gestational VE change per week						
<0		**0.067 (0.052, 0.083)**	**0.050 (0.019, 0.082)**		**1.16 (1.08, 1.25)**	1.09 (0.94, 1.26)
0 to 0.19		0.00 [Reference]	0.00 [Reference]		1.00 [Reference]	1.00 [Reference]
0.20 to 0.29		**−0.014 (−0.027, 0.000)**	0.000 (−0.026, 0.026)		1.01 (0.94, 1.08)	0.99 (0.87, 1.12)
≥0.30		**−0.033 (−0.045, −0.021)**	**−0.040 (−0.063, −0.017)**		**0.93 (0.87, 0.98)**	0.94 (0.83, 1.06)

We used a generalized linear model (GLM) to assess the association of the first-trimester VE concentration categories and gestational VE change categories with FPG and used robust Poisson regression to assess the association with GDM. Model A was adjusted for region, household registration, ethnic origin, education, age, ART, primigravida, pre pregnancy BMI, hypertension, and FPG levels at gestational age of 4–13 weeks. Model B was additionally adjusted for the confounding of the first-trimester VE concentration categories and gestational VE change categories on each other. Model C includes an interaction term of the first-trimester VE concentration categories and gestational VE change categories based on Model B, additionally. Bold number indicates the β or RR (95% Cis) were statistically significant with *p*-value < 0.05.

**Table 3 nutrients-15-01598-t003:** Adjusted relative risks (95% CI) for GDM according to gestational vitamin E (VE) changes stratified by the first-trimester VE concentration categories.

	FPG Concentration Differences across Relative Change Categories [β (95% CI)]				GDM Risk Differences across Relative Change Categories [RR (95% CI)]			
	<0	0 to 0.19	0.20 to 0.29	≥0.30	<0	0 to 0.19	0.20 to 0.29	≥0.30
Serum vitamin E concentrations during 4–13 week (mg/L)								
<7.2	**0.162 (0.054, 0.271)**	0.00 [Reference]	−0.071 (−0.148, 0.007)	−0.054 (−0.119, 0.011)	**1.52 (1.12, 2.07)**	1.00 [Reference]	0.90 (0.68, 1.20)	0.96 (0.77, 1.19)
7.2 to 7.9	**0.314 (0.156, 0.472)**	0.00 [Reference]	0.009 (−0.054, 0.072)	−0.016 (−0.071, 0.039)	**2.47 (1.35, 4.52)**	1.00 [Reference]	1.18 (0.84, 1.65)	1.03 (0.76, 1.38)
8.0 to 9.3	**0.257 (0.182, 0.333)**	0.00 [Reference]	0.021 (−0.010, 0.052)	−0.012 (−0.040, 0.016)	1.28 (0.93, 1.76)	1.00 [Reference]	1.07 (0.91, 1.25)	**0.86 (0.74, 0.99)**
9.4 to 11.0	**0.157 (0.114, 0.200)**	0.00 [Reference]	−0.012 (−0.036, 0.012)	−0.018 (−0.039, 0.003)	**1.45 (1.20, 1.76)**	1.00 [Reference]	1.00 (0.88, 1.14)	0.89 (0.79, 1.00)
11.1 to 13.2	**0.049 (0.018, 0.079)**	0.00 [Reference]	−0.005 (−0.031, 0.020)	**−0.045 (−0.067, −0.022)**	1.09 (0.94, 1.27)	1.00 [Reference]	0.97 (0.86, 1.11)	0.92 (0.82, 1.04)
13.3 to 15.8	0.028 (−0.003, 0.060)	0.00 [Reference]	**−0.073 (−0.113, −0.032)**	**−0.053 (−0.090, −0.017)**	1.11 (0.96, 1.28)	1.00 [Reference]	0.90 (0.74, 1.10)	0.95 (0.80, 1.14)
15.9 to 17.7	0.026 (−0.021, 0.072)	0.00 [Reference]	0.005 (−0.085, 0.095)	−0.023 (−0.099, 0.054)	1.13 (0.89, 1.44)	1.00 [Reference]	1.32 (0.85, 2.05)	1.05 (0.70, 1.59)
17.8 to 35.9	0.019 (−0.039, 0.076)	0.00 [Reference]	0.074 (−0.066, 0.213)	−0.051 (−0.175, 0.074)	0.99 (0.77, 1.29)	1.00 [Reference]	1.46 (0.83, 2.57)	1.17 (0.65, 2.08)

We performed a stratified analysis to examine the association of gestational VE change categories with FPG and GDM in each first-trimester VE concentration cate-gory, by using robust Poisson regression models that adjusted for region, household registration, ethnic origin, education, age, ART, primigravida, pre pregnancy BMI, and hypertension. Bold number indicates the β or RR (95% Cis) were statistically significant with *p*-value < 0.05.

## Data Availability

The datasets generated during and/or analyzed during the current study are available from the corresponding author upon reasonable request.

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
