# Peer review of "Gestational Vitamin E Status and Gestational Diabetes Mellitus: A Retrospective Cohort Study"

_nutrients, 2023, doi:10.3390/nu15071598_

Round 1
Reviewer 1 Report
REVIEW of the study by Shi H, Gong X, Sheng Q, Li X, Wang Y, Wu T, Zhao Y, and Wei Y. Gestational vitamin E status and gestational diabetes mellitus (GDM): a retrospective study.
Though the subject is not entirely new [e.g. Maktabi M et al. Lipids Health Dis 2018; 17 (1):163] this a very large (n=52 791 participants) and multicentre study (180 maternity units) that addresses a relationship between vitamin E status and the risk of GDM. The authors conclude that there is a L-shape relationship between the risk of GDM and vitamin E status and the risk of GDR (RR 1.36), particularly for those with 1sta trimester vitamin E below 7.2 mg/dl. Also failure of pregnancy-related to increase in vitamin E was also associated with an increased risk of GDM.
These observations are interesting and in vie of a very large investigated cohort, in my opinion, merit publication.
There are, however, certain aspects of the paper that should be improved and discussed in greater detail.
- The authors used only fasting glucose to diagnose GDM with a cut-off point similar to IADPSG diagnostic GDM criteria, but they did not use an oral glucose tolerance test. Though this is shortly mentioned as a limitation of the study (lines 269-271), the potential impact of this drawback is not adequately discussed. It is well recognised that fasting hyperglycaemia represents predominantly hepatic insulin resistance, while post-prandial glucose excursions (and glucose excursions during OGTT) represent predominantly peripheral insulin resistance [e.g. Hoffman RP. Indices of insulin action calculated from fasting glucose and insulin reflect hepatic, not peripheral, insulin sensitivity in African-American and Caucasian adolescents. Pediatr Diabetes 2008; 9: 57-61]. Furthermore, correlation between insulin resistance indices derived from fasting values and during OGTT is positive but relatively modest and this is also true for pregnancy, potentially influencing the prevalence of GDM according to the criteria used [e.g. discussed in Lewandowski K et al. Endokrynol Pol 2022;73(1):1-7. DOI: 10.5603/EP.a2021.0095]. Hence, application of an OGTT for diagnosis of GDM, rather than just fasting glucose, might have significantly altered some conclusions of the study, or at least affect some the calculated cut-off points for vitamin E concentrations associated with an increased risk of GDM. The significance of this should be discussed in more detail.
- The authors report surprisingly high prevalence of an early (i.e. first trimester) GDM (from 12.64% up to 22.44%, lines 150-152), but do not compare their findings to other studies, where prevalence of early GDM was investigated. Is there an issue for universal screening for an early GDM in China?
- I would be cautious in suggesting vitamin E supplementations at this point, when dietary advice should be offered first (lines 257-258).
- I have found in Wikipedia what “Hukou” means (line 109), and it apparently applies to Chinese population registration system, but such terms should be explained to foreign readers.
Author Response
Comments and Suggestions for Authors:
REVIEW of the study by Shi H, Gong X, Sheng Q, Li X, Wang Y, Wu T, Zhao Y, and Wei Y. Gestational vitamin E status and gestational diabetes mellitus (GDM): a retrospective study.
Though the subject is not entirely new [e.g. Maktabi M et al. Lipids Health Dis 2018; 17 (1):163] this a very large (n=52 791 participants) and multicentre study (180 maternity units) that addresses a relationship between vitamin E status and the risk of GDM. The authors conclude that there is a L-shape relationship between the risk of GDM and vitamin E status and the risk of GDR (RR 1.36), particularly for those with 1sta trimester vitamin E below 7.2 mg/dl. Also failure of pregnancy-related to increase in vitamin E was also associated with an increased risk of GDM.
These observations are interesting and in vie of a very large investigated cohort, in my opinion, merit publication.
There are, however, certain aspects of the paper that should be improved and discussed in greater detail.
Point 1: The authors used only fasting glucose to diagnose GDM with a cut-off point similar to IADPSG diagnostic GDM criteria, but they did not use an oral glucose tolerance test. Though this is shortly mentioned as a limitation of the study (lines 269-271), the potential impact of this drawback is not adequately discussed. It is well recognised that fasting hyperglycaemia represents predominantly hepatic insulin resistance, while post-prandial glucose excursions (and glucose excursions during OGTT) represent predominantly peripheral insulin resistance [e.g. Hoffman RP. Indices of insulin action calculated from fasting glucose and insulin reflect hepatic, not peripheral, insulin sensitivity in African-American and Caucasian adolescents. Pediatr Diabetes 2008; 9: 57-61]. Furthermore, correlation between insulin resistance indices derived from fasting values and during OGTT is positive but relatively modest and this is also true for pregnancy, potentially influencing the prevalence of GDM according to the criteria used [e.g. discussed in Lewandowski K et al. Endokrynol Pol 2022;73(1):1-7. DOI: 10.5603/EP.a2021.0095]. Hence, application of an OGTT for diagnosis of GDM, rather than just fasting glucose, might have significantly altered some conclusions of the study, or at least affect some the calculated cut-off points for vitamin E concentrations associated with an increased risk of GDM. The significance of this should be discussed in more detail.
Response 1: Thanks for your advice. We agree with that the application of an OGTT for diagnosis of GDM might affact the results of the study, which is also a limitation for this study. Our retrospective data were collected from 180 maternity hospitals from most part of China, including low-resource Settings where not all pregnanct women had opportunity to conduct OGTT. Hence, we used FPG of ≥ 5.1 mmol/l as the criteria.
According to your advice, we have modified the discussion. “However, our research has several limitations. We diagnosed GDM by FPG in 24-40 weeks. It is well recognized that fasting hyperglycaemia represents predominantly hepatic insu-lin resistance, while post-prandial glucose excursions (and glucose excursions during OGTT) represent predominantly peripheral insulin resistance[25]. Hence, application of an OGTT for diagnosis of GDM, rather than just using FPG, might alter the result of the study, or at least affect some the calculated cut-off points for VE concentrations associated with an increased risk of GDM. However, our retrospective data were collected from 137 maternity hospitals from 22 provinces of China, including low-resource Settings where not all pregnanct women have available access to OGTT. According to the IADPSG criteria, FPG of 4.8-5.0 mmol/l showed good test characteristics in screening for GDM[26]. Inter-estingly, another study found that mild hyperglycemia (FPG of 5.1-5.5 mmol/l) was more related with the increased risk of large for gestational age than GDM diagnosis by OGTT [27].”
Point 2: The authors report surprisingly high prevalence of an early (i.e. first trimester) GDM (from 12.64% up to 22.44%, lines 150-152), but do not compare their findings to other studies, where prevalence of early GDM was investigated. Is there an issue for universal screening for an early GDM in China?
Response 2: Thanks for your remingding. In this study, the prevalence of GDM was 13.57%, which was similar as the estimate reported by previous study (14.8%). However, the prevalence, 12.64% up to 22.44%, was calculated by the concentrations of vitamin E, which was 7.2 (<5th), 7.2–7.9 (5th–9th), 8.0–9.3 (10th–24th), 9.4–11.0 (25th–49th),11.1–13.2 (50th–74th), 13.3–15.8 (75th–89th), 15.9–17.7 (90th–95th), and 17.8–35.9 (95th–100th) mg/L in the first trimester, respectively.
Point 3: I would be cautious in suggesting vitamin E supplementations at this point, when dietary advice should be offered first (lines 257-258).
Response 3: Thanks for your advice. We revised the sentences as “These findings suggest the necessity of having foods rich in vitamin E and appropriate vitamin E supplementation to prevent GDM for pregnant women with low baseline vitamin E levels, although the idea needs further confirmation”.
Point 4: I have found in Wikipedia what “Hukou” means (line 109), and it apparently applies to Chinese population registration system, but such terms should be explained to foreign readers.
Response 4: Thanks for your advice. “Hukou” has been used in previous paper, which applied to Chinese population registration system. We have changed it to “Household registration”.

Reviewer 2 Report
The authors describe the association of vitamin E with fasting plasma glucose and with the rate of GDM diagnosed based on fasting plasma glucose. The study included an impressive number of participants. However, it is strange that being able to perform a more sophisticated analysis of vitamin E, they were not able to perform the oral glucose tolerance test.
Major comment:
1. It is unclear why the authors decided to study this association. Vitamin E has implications in pathogenesis of diabetes complications, but the evidence of its role in the development of hyperglycemia is lacking.
The authors claim that «Oxidative stress may be implicated in the pathogenesis of diabetes mellitus...[15] However, ref 15 concerns only diabetes complications, but not the pathogenesis of diabetes.
Please provide some hypothesis on the role of Vitamin E in the the pathogenesis of GDM.
2. Ref 18 «Analysis of correlation between vitamin A, E and gestational diabetes mellitus» has no doi and is not available in the internet. Judging by the title, it describes only correlation, but not the effect of supplementation of vitamin E during pregnancy on reducing the risk of GDM.
The potential application value of the study is very questionable because Serum vitamin E concentrations very close to 7 (7.2 to 7.9) were associated with the decreased risk of GDM (0.87 (0.76, 0.98) compared to the reference range of 11.1 to 13.2 (table 2).
Minor comments
Line 87 – 88 The meaning of the phrase is unclear: «we extracted the minimum vitamin E concentration … with the highest FPG value in 24-40 weeks». What do you mean under «extracted»?
Line 97 The meaning of the phrase is unclear: «We extracted the highest fasting plasma glucose values at gestational age of 4–13». Does it mean that the women in the study had multiple measurements of plasma glucose in the first trimester?
Table 3. What is «Total2»?
Author Response
The authors describe the association of vitamin E with fasting plasma glucose and with the rate of GDM diagnosed based on fasting plasma glucose. The study included an impressive number of participants. However, it is strange that being able to perform a more sophisticated analysis of vitamin E, they were not able to perform the oral glucose tolerance test.
Response 1: Thanks for your comments. We agree with that the application of an OGTT for diagnosis of GDM might affact the results of the study, which is also a limitation for this study. Our retrospective data were collected from 180 maternity hospitals across 23 provinces of China, including low-resource Settings where not all pregnanct women had opportunity to conduct OGTT. Hence, we used FPG of ≥ 5.1 mmol/l as the criteria.
Major comment:
Point 1: It is unclear why the authors decided to study this association. Vitamin E has implications in pathogenesis of diabetes complications, but the evidence of its role in the development of hyperglycemia is lacking.
The authors claim that «Oxidative stress may be implicated in the pathogenesis of diabetes mellitus...[15] However, ref 15 concerns only diabetes complications, but not the pathogenesis of diabetes.
Please provide some hypothesis on the role of Vitamin E in the the pathogenesis of GDM.
Response 1: Thanks for your advice. In the revised version, we restated why we did the study and added an explanation on the potential role of Vitamin E in the the pathogenesis of GDM. “Oxidative stress may be implicated in the pathogenesis of diabetes mellitus and its complications[4-8]. The underlying pathophysiology is similar to type 2 diabetes in most instances, which is hyperglycemia leads to excessive production of intracellular reactive oxygen species (ROS) with insufficient clearance, thus enhancing the oxidative stress level of the body, leading to adverse pregnancy outcomes[4-7]. Vitamin E (VE), a well-documented antioxidant, can neutralize ROS, eliminate oxygen free radicals and protect tissue from damage[9,10]. However, although a robust number of studies on the role of VE in the cause and prevention of diabetic complications[8,11-14], there are very few studies on its association with risk of diabetes mellitus, especially GDM. The system-atic review by Sharifipour et al aimed to evaluate the association between VE and GDM, and only included eleven relevant studies with a total of 596 participants published before December 2019, of which seven were the case-control studies, and the rest were cross-sectional[15]. Our further literature search identified very few relevant articles pub-lished since then. The biggest hurdle is the cost and labor involved in obtaining a large enough sample size to examine the quantitative association, given the low proportion of VE deficiency. More researches are evidently needed on this topic, especially those with longitudinal design. In response to the lack of research, we did this retrospective large-sample cohort study to examine the association between VE and GDM.”
Point 2: Ref 18 «Analysis of correlation between vitamin A, E and gestational diabetes mellitus» has no doi and is not available in the internet. Judging by the title, it describes only correlation, but not the effect of supplementation of vitamin E during pregnancy on reducing the risk of GDM.
Response 2: Thanks for your advice. Ref 18 is a paper from Chinese journal and reported a retrospective study. considering it is not an intervention study, we deleted it according to your advice.
In addtion, all references’s doi were supplemented except for those without doi.
Point 3: The potential application value of the study is very questionable because Serum vitamin E concentrations very close to 7 (7.2 to 7.9) were associated with the decreased risk of GDM (0.87 (0.76, 0.98) compared to the reference range of 11.1 to 13.2 (table 2).
Response 3: This result (0.87 (0.76, 0.98) comes from Model A, but a more accurate result should be seen in Model C which adjusted for age, education, ethnic origin, region, Household registration, ART, primigravida, pre pregnancy BMI, hypertension, and FPG levels at gestational age of 4–13 weeks, the confounding of the vitamin E concentrations categories in the first trimester and the gestational change categories on each other, and an interaction term of the vitamin E concentrations categories in the first trimester and the gestational change categories. In model C, there was no significant difference in FGP concentration or GDM risk between women with serum vitamin E concentrations of 7 (7.2 to 7.9) mg/L during 4-13 week, compared to the reference range of 11.1 to 13.2.
Minor comments
Point 1: Line 87 – 88 The meaning of the phrase is unclear: «we extracted the minimum vitamin E concentration … with the highest FPG value in 24-40 weeks». What do you mean under «extracted»?
Response 1: We revised the sentence as the follow: we used the minimum vitamin E concentration tested in the first trimester and the vitamin E concentration at the time of FPG test with the highest FPG value in 24-40 weeks.
Point 2: Line 97 The meaning of the phrase is unclear: «We extracted the highest fasting plasma glucose values at gestational age of 4–13». Does it mean that the women in the study had multiple measurements of plasma glucose in the first trimester?
Response 2: Yes, a few pregnant women may have multiple measurements during the first trimester and during 24-40 weeks of pregnancy
Point 3: Table 3. What is «Total2»?
Response 3: Sorry for that. We already deleted total1 and total2.

Round 2
Reviewer 2 Report
The manuscript has been sufficiently improved to warrant publication in Nutrients.
Author Response
Thank you very much for approving our revision.